# MSF Enhances Human Antimicrobial Peptide *β*-Defensin (HBD2 and HBD3) Expression and Attenuates Inflammation via the NF-*κ*B and p38 Signaling Pathways

**DOI:** 10.3390/molecules28062744

**Published:** 2023-03-18

**Authors:** Anh-Thu Nguyen, Minho Kim, Ye-Eun Kim, Hangeun Kim, Sanghyun Lee, Yunji Lee, Ki-Young Kim

**Affiliations:** 1Department of Genetics and Biotechnology, Kyung Hee University, Youngin 1732, Republic of Korea; 2Research and Development Center, Skin Biotechnology Center Co., Ltd., Yongin 17104, Republic of Korea; 3Department of Plant Science and Technology, Chung-Ang University, Anseong 17546, Republic of Korea; 4Department of Herbal Crop Research, National Institute of Horticultural and Herbal Science, Eumseong 27709, Republic of Korea

**Keywords:** MSF, *L. plantarum K8*, *F. glaberrima* Nakai, defensin, anti-inflammation

## Abstract

Both defensin and inflammation are part of the human innate immune system that responds rapidly to pathogens. The combination of defensins with pro- or anti-inflammatory effects can be a potential research direction for the treatment of infection by pathogens. This study aimed to identify whether MSF (Miracle Synergy material made using *Filipendula glaberrima*)*,* a probiotic lysate of *Filipendula glaberrima* extracts fermented with *Lactiplantibacillus plantarum* K8, activates the expression of human *β*-defensin (HBD2 and HBD3) to protect the host against pathogens and inhibit inflammation caused by *S. aureus*, in vitro with Western blot analysis, qRT-PCR and in vivo studies with a mouse model were used to evaluate the effects of MSF. The MSF treatment induced HBD2 and HBD3 expression via the p38 and NF-*κ*B pathways. Furthermore, MSF treatment significantly reduced the expression of pro-inflammatory cytokines (TNF-*α*, IL-1*β*, IL-6, and IL-8), also through p38 and NF-*κ*B in *S. aureus*-induced inflammatory condition. MSF treatment remarkably reduced erythema in mice ears caused by the injection of *S. aureus*, while K8 lysate treatment did not initiate a strong recovery. Taken together, MSF induced the expression of HBD2 and HDB3 and activated anti-inflammatory activity more than the probiotic lysates of *L. plantarum* K8. These findings show that MSF is a potential defensin inducer and anti-inflammatory agent.

## 1. Introduction

Skin is the largest organ of the body, and acts as the primary barrier against microbial infection [1]. The protection mechanism of the skin includes the physical representation of the epidermis, stratum corneum, antimicrobial peptides (AMPs), and the physiological representation of innate immune responses [1,2,3]. While barrier defenses are the body’s first line of physical defense systems, innate immune responses are the first line of physiological defense systems against pathogens [1,3]. Indeed, both mechanisms can cooperate, and one can stimulate the other in their responses against pathogens.

Human defensins work as a part of the innate immune system and are a major family of AMPs expressed predominantly in epithelial cells and neutrophils [4]. Human defensins consist of two genetically distinct forms, *α*- and *β*-defensins [5]. Human *α*-defensins are mainly found in granules of polymorphonuclear leukocytes (HNP1-4) and in small intestinal Paneth cells (HBD5 and HBD6), whereas human *β*-defensins (HBD1-4) are found in leukocytes and epithelial cells [5,6]. HBD1 is constitutively activated, while HBD2 and HBD3 are inducible in the presence of a variety of stimuli including infections [5]. Depending on the pathogens and disease progress, defensins can combine pro- and anti-inflammatory effects.

Inflammation is an innate immune response that protects the body against harmful stimuli such as damaged cells, pathogens, or irritants [7]. Inflammation is initiated in response to substances (peptides, pro-inflammatory cytokines, and chemokines), which are secreted by leukocytes, to mediate the inflammatory process [7]. Commercially available anti-inflammatory medicines are associated with adverse effects to varying degrees. Hence, anti-inflammatory medicines with low toxicity and good efficacy would be very beneficial [8]. Natural resources, such as herbal plants and microorganisms, are potential candidates for anti-inflammatory medicines.

In this study, MSF (Miracle Synergy material made using *F. glaberrima*), a probiotic lysate of *F. glaberrima* extracts (FGE) fermented with *L. plantarum* K8 to increase FGE biological activity was tested [9]. MSF water extract is a product using miracle synergy material that can completely eliminate the risk of sediment or microbial growth that occurs during manufacturing. It provides higher efficacy than the simple filtration method used to apply existing lactic acid bacteria lysates to products. *L. plantarum* (formerly *Lactobacillus plantarum*) is an aerobic bacterium belonging to Gram-positive lactic acid bacteria species [9,10]. *L. plantarum* produces antimicrobial peptides that enhance the antimicrobial activities of the skin and can act as an anti-inflammatory substance. *L. plantarum* is an interesting probiotic candidate inhabiting a range of ecological niches including fermented foods, meats, and plants [10]. The *L. plantarum* K8 strain was originally isolated from kimchi and is particularly active in sour kimchi [9,10]. *F. glaberrima* Nakai (also called Korean meadowsweet) is a perennial herbaceous plant in the Rosaceae family and is found in the mountains of central Korea [11]. Throughout history, people have been using *F. glaberrima* Nakai as a remedy to treat inflammation, pain, and gout [11].

Accordingly, MSF was tested for antimicrobial activity and anti-inflammatory activity.

## 2. Results

### 2.1. MSF Significantly Enhanced the Expression of Human β-Defensin-2 and β-Defensin-3

To test whether MSF induces the expression of genes encoding AMPs (HBD1-3), MSF in concentrations of 25–100 µg/mL was used to treat human keratinocyte cells (HaCaT) for periods of 1–48 h. K8 lysate was used as a control.

Analysis with qRT-PCR showed that MSF significantly induced the expression of HBD2 and HBD3 in a time- and dose-dependent manner (Figure 1A,B), but the expression of HBD1 was time-dependently decreased (Figure 1A) and dose-dependently increased (Figure 1B). MSF treatment increased HBD2 and HBD3 expressions, more than K8 lysate treatment at the same concentration.

The expression of HBD2 and HBD3 protein levels was also induced by treatment with the indicated concentration of MSF (Figure 1C). MSF should be able to increase the antimicrobial activity of cells against pathogenic bacteria.

### 2.2. Effect of MSF on the PI3K, NF-κB, and MAPKs Signaling Pathways in HaCaT Cells

PI3K, NF-*κ*B, and MAPKs signaling pathways have a critical role in innate and adaptive immunity. Class I PI3Ks activate and elicit cellular responses of each cell that expresses receptors for immune responses [12]. The NF-*κ*B and MAPK pathways are key for epithelial immune defense and have been implicated in the secretion of antimicrobial peptides, the release of cytokines/chemokines to mobilize immune effector cells, and the activation of adaptive immunity [13]. Therefore, to determine which pathway among the PI3K, NF-*κ*B, and MAPKs signaling pathways are involved in the regulation of the expression of HBD1, HBD2, and HBD3, HaCaT cells were treated with 25–100 µg/mL of MSF.

The phosphorylation of NF-*κ*B, p38, and ERK1/2 was time- and dose-dependently induced by treatment with MSF or K8 lysate (Figure 2A,B). However, ERK1/2 phosphorylation increased with a dose of 100 µg/mL within 1 h of treatment and decreased after that.

In addition, the expression of genes encoding pro-inflammatory mediators such as TNF-*α*, IL-1*β*, IL-6, and IL-8, which are generally considered to respond together with antimicrobial peptide genes during the innate immune response, were also investigated. The expression of TNF-*α*, IL-1*β*, IL-6, and IL-8 increased slightly in a dose-dependent manner by treatment with MSF or K8 lysate (Appendix A). However, the expression of IL-1*β*, IL-6, and IL-8 was time-dependently reduced by treatment with MSF or K8 lysate, but the expression of TNF-*α* was induced by the MSF or K8 lysate treatment (Appendix A).

Assay with MTT consistently showed that the viability of HaCaT cells was not affected by MSF or K8 lysate treatment up to a concentration of 100 µg/mL (Figure 2C).

Collectively, treatment with MSF induced the expression of *β*-defensin (HBD2 and HBD3) and affected the expression of pro-inflammatory genes (TNF-*α*, IL-1*β*, IL-6, and IL-8), but it was not clear which pathways, including the p38 and the NF-*κ*B pathways were involved in increasing the expression.

### 2.3. MSF Promoted the Expression of HBD2 and HBD3 through p38 Pathway

To verify whether p38 mediated the HBD2 and HBD3 expression caused by the MSF treatment, SB203580, a specific inhibitor of p38, was used. In the presence of SB203580, the expression of HBD2 and HBD3 did not increase by treatment with 25, 50, and 100 µg/mL of MSF or K8 lysate (Figure 3A,B). However, a slightly reduced expression of pro-inflammatory cytokines (TNF-*α*, IL-1*β*, IL-6, and IL-8) was shown by treatment with MSF or K8 lysate (Appendix A compared to Appendix A). These results suggested that MSF induced the expression of HBD2 and HBD3 through the p38 pathway.

### 2.4. NF-κB Involved in the Induction of HBD3 and HBD2 Expression

To examine whether MSF treatment increases HBD2 and HBD3 expression through the NF-*κ*B pathway, HaCaT cells were pre-treated for 3 h with SP100030, an NF-*κ*B specific inhibitor, and then treated with 25–100 µg/mL of MSF or K8 lysate for 1 h. The phosphorylated form of NF-*κ*B was not induced by treatment with MSF or K8 lysate after the treatment with SP100030 (Figure 4A compared to Figure 2B). Inhibition of NF-*κ*B decreased the expression of HBD2 and HBD3 by treatment with MSF and K8 lysate (Figure 4B). Furthermore, the expression of pro-inflammatory genes (TNF-*α*, IL-1*β*, IL-6, and IL-8) decreased slightly depending on the concentration of the MSF treatment (Appendix A compared to Appendix A). Overall, MSF treatment increased the expression of HBD2 and HBD3 via the NF-*κ*B pathway.

### 2.5. MSF Inhibited S. aureus-Induced Inflammation in THP1 Cells

*Staphylococcus aureus (S. aureus)* infections play an extremely important role in a variety of diseases including inflammation diseases. *S. aureus* triggers major intracellular signaling pathways: MAPK pathways including p38, ERK1/2, and JNK, and the pathway leading to activation of the transcription factor NF-*κ*B. To test the anti-inflammatory effects of MSF and K8 lysate in THP1 cells, THP1 cells were pre-treated with 25–100 µg/mL of MSF or K8 lysate for 1 h, and then cultured with heat-killed *S. aureus* for 24 h. The treatment of heat-killed *S. aureus* increased the expression of pro-inflammatory cytokines including TNF-*α*, IL-1*β*, IL-6, and IL-8, while treatment with MSF or K8 lysate significantly inhibited the gene expression of the pro-inflammatory cytokines to a similar extent in a dose-dependent manner (Appendix A and Figure 5A).

The MAPKs, including p38, ERK1/2, and JNK, are involved in the signal transduction pathways and NF-*κ*B is critically required for the transcriptional regulation of genes for inflammation [14,15,16,17]. Accordingly, the inflammatory response after MSF treatment through the MAPKs or NF-*κ*B pathway in *S. aureus*-treated THP1 cells was tested by Western blot analysis. The treatment of *S. aureus* induced the phosphorylation of NF-*κ*B and p38, but not ERK1/2 and JNK. MSF reduced NF-*κ*B and p38 phosphorylation which was increased by the treatment with *S. aureus* (Figure 5B).

In addition, MSF and K8 lysate did not show any toxicity to the THP1 cells (Figure 5C).

Therefore, MSF showed anti-inflammatory effects by suppressing the expression of pro-inflammatory cytokines (TNF-*α*, IL-1*β*, IL-6, and IL-8) through the NF-*κ*B and p38 pathways.

### 2.6. TLR2 Was Involved in Regulating Human β-Defensin and Inflammation of MSF

TLR2 and TLR4 play crucial roles in modulating inflammatory response by recognizing different bacterial cell wall components, then activating the transcription factors via intracellular pathways, and generating cytokines and chemokines [18,19].

The expression of TLR2 and TLR4 were tested after treatment with 25–100 µg/mL MSF or K8 lysate to determine to check whether compounds influence the ability of TLR2 and TLR4 to regulate the expression of HBD2, HBD3, and pro-inflammation cytokines. TLR2 expression was increased by treatments with either MSF or K8 lysate, while TLR4 expression decreased (Figure 6A). In addition, the mRNA gene expression of TLR2 and TLR4 was reduced by MSF or K8 lysate treatment in *S. aureus*-stimulated THP1 cells (Figure 6B) and similar results with protein levels (Figure 6C). These results suggested that TLR2 plays a role in regulating the expression of defensin and inflammatory cytokines by MSF treatment.

### 2.7. MSF Ameliorated S. Aureus-Induced Skin Inflammation in Mice

The in vivo pathophysiological effect of MSF using a BALB/cAnNTac mouse model was evaluated (Figure 7A). To achieve *S. aureus*-induced inflammation in the mouse model, 10^8^ colony-forming units (CFUs) of living *S. aureus* were intradermally injected into the ears of the mouse. After 24 h of *S. aureus* administration, the ears exhibited significant cutaneous erythema and swelling, a typical symptom of ear inflammation (Figure 7B, right ear). MSF treatment remarkably reduced erythema comparable to no treatment (*S. aureus* injection only), while K8 lysate treatment did not show a strong recovery (Figure 7B, left ear).

To verify the reduction in ear inflammation by MSF treatment, the expression of pro-inflammatory cytokines such as TNF-*α*, IL-1*β*, IL-6, and IL-8 was tested using qRT-PCR. *S. aureus* injection increased the expression of TNF-*α*, but treatment with MSF significantly decreased the expression of TNF-*α* (Figure 7C). The expression of IL-1*β* and IL-6 was not decreased when treated with MSF in the *S. aureus*-stimulated left mouse ear (Figure 7C). IL-8 was undetectable. Together, the data demonstrate that MSF reduced the inflammatory response caused by *S. aureus* injection in mice ears.

## 3. Discussion

Defensins are effector molecules of the innate host defense system with antimicrobial activity against a variety of pathogens, including *S.aureus* [5]. Human *β*-defensin is an important factor to induce or regulate the host defense and may bridge innate and adaptive immunity at surfaces [5].

*F. glaberrima* Nakai has been used as a remedy to treat inflammation, pain, and gout and the ethanol extract of *F. glaberrima* Nakai on anti-inflammatory activity has been reported [11]. In addition, *L. plantarum* K8 lysate has shown anti-inflammatory effects and alleviated lipopolysaccharide (LPS)-induced septic shock [9].

Due to this, we hypothesized that MSF, a fermented product of *F. glaberrima* Nakai with *L. plantarum* K8, was a potential candidate to be a defensin inducer and an anti-inflammatory agent. To explore these possibilities, the expressions of *β*-defensin HBD1, HBD2, and HBD3 were tested after treatment with MSF. MSF treatment induced the expression of the mRNA and protein levels of HBD2 and HBD3 (Figure 1). Three important pathways (PI3K, NF-*κ*B, and MAPKs) in the inflammatory response were concurrently checked to determine which participated in the regulation of the expression of defensin and inflammatory cytokines after treatment with MSF. MSF treatment activated HBD2 and HBD3 expression through the NF-*κ*B and p38 pathways (Figure 1 and Figure 2). Early responses indicated that NF-*κ*B and p38 were activated after only 1 h treatment, but not PI3K, AKT, and JNK signaling when HaCaT cells were treated with MSF or K8 lysate. ERK1/2 was also activated but only for the first 1 h and decreased thereafter, so we did not check ERK1/2 in this study.

The MAPK signaling pathway has important roles in innate immune responses, ranging from the induction of pro-inflammatory mediators, such as cytokines and chemokines, to the activation of anti-inflammatory feedback pathways [14,15]. In mammalian cells, there are three well-defined MAPK pathways: the extracellular-signal-regulated kinase (ERK) pathway, the JUN N-terminal kinase (JNK) pathway, and the p38 pathway [14,15]. p38 is protective in multiple autoimmune and inflammatory. The mRNA and protein expression of HBD2 and HBD3 by MSF or K8 lysate treatment were mediated through p38 signaling (Figure 1 and Figure 2). Inhibition of the p38 pathway by SB203580 blocked the induction of HBD2 and HBD3, confirming the central role of p38 in the regulation of HBD2 and HBD3 expression (Figure 3).

The transcription factor NF-*κ*B regulates the expression of a large array of genes involved in processes of the immune and adaptive immune functions and serves as a pivotal mediator of inflammatory responses [16,17]. NF-*κ*B has long been considered a prototypical pro-inflammatory signaling pathway, largely based on the induction of the expression of several pro-inflammatory genes, including those genes encoding cytokines, chemokines, and inflammasome regulation [16,17]. Treatment of MSF or K8 lysate with NF-*κ*B inhibitor showed that NF-*κ*B enhanced HBD2, HBD3, confirming NF-*κ*B regulated HBD2 and HBD3 expression (compare Figure 1, Figure 2, and Figure 4).

Taken together, MSF activated both NF-*κ*B and p38 pathways, all p38 and NF-*κ*B induced the expression of the HBD2 and HBD3. All these results showed that MSF possessed antimicrobial activity and could provide the first line of defense in the immune system.

On another hand, treatment with MSF influenced the expression of TNF-*α*, IL-1*β*, IL-6, and IL-8 via p38 and NF-*κ*B signaling pathways (Appendix A). The differential transcriptional activation of pro-inflammatory genes is precisely controlled by the selective binding of transcription factors to the promoter of these genes. Furthermore, many signaling pathways regulate the expression of pro-inflammatory cytokine genes. NF-*κ*B, p38, and ERK1/2 are upstream signaling of TNF-*α*, IL-1*β*, IL-6, and IL-8. Therefore, the different expressions of NF-*κ*B, p38, and ERK1/2 lead to the different expression patterns of TNF-*α*, IL-1*β*, IL-6, and IL-8.

In addition, *S. aureus* induced pro-inflammatory gene expression in THP1 cells, while MSF reduced inflammatory responses caused by *S. aureus* (Appendix A and Figure 5). Inhibition of the signaling cascade including p38 or NF-*κ*B by MSF could lead to suppression of inflammatory gene expression in *S. aureus*-stimulated THP1 cells with higher anti-inflammatory efficacy than conventional probiotic lysates (K8 lysate). However, compared with the previously reported anti-inflammatory response of MSF [9], the different experimental conditions led to a slightly different pattern of gene expression. Therefore, we can suggest that MSF has a potential anti-inflammatory activity.

The toll-like receptor is a pattern recognition receptor that has a key role in enabling cells of the innate immune system to recognize the conserved structural motifs on a wide array of pathogens including Gram-positive bacteria [18,19]. The treatment of MSF increased the expression of TLR2, but not TRL4 (Figure 6) suggesting that MSF activated TLR2 leading to the expression of pro-inflammatory cytokines such as TNF-*α*, IL-1*β*, IL-6, and IL-8 through the p38 or NF-*κ*B pathways.

To determine the physiological effect of MSF on the *S. aureus* infection condition, we used a well-known mouse model. Injection of *S. aureus* increased the relative ratio of ear swelling, but treatment with 100 µg/mL MSF significantly decreased ear swelling (Figure 7B). Furthermore, the expression of TNF-*α* was reduced (Figure 7C), but IL-6 and IL-1*β* in the ears of mice treated with MSF are higher than that in the other ear. In this experiment, while the mice’s ears were sampled for RT-PCR, the tissues included various cell types in the skin that produced cytokines. It is possible that the reductions or inductions in cytokine expression could occur in these cell types as well. Additionally, the IL-6 and IL-1*β* gene expression might be less sensitive to MSF inhibition *S. aureus*-induced inflammation compared to TNF-*α* in vivo model. Alternatively, it could be that IL-6 and IL-1*β* downregulation by MSF in our mice is regulated by other pathways. These results suggested that MSF ameliorated *S. aureus*-induced inflammation by reducing the expression of cytokines.

Based on these results, we proposed a possible mechanism for how MSF induces the expression of human *β*-defensins and inhibits the inflammation caused by *S. aureus* (Figure 8).

## 4. Materials and Methods

### 4.1. Preparation of MSF

MSF was prepared based on a previous report by culturing *L. plantarum* K8 in De Man, Rogosa, and Sharpe (MRS) broth (BD Bioscience, San Jose, CA, USA), containing *F. glaberrima* Nakai extracts [9]. Cultivation was carried out by inoculating the prepared extract of *F. glaberrima* Nakai into the inoculum of *L. plantarum* K8 at 37 °C overnight and recovering the lactic acid bacteria cells using a continuous centrifugal separator. Bacteria were re-suspended in deionized water and disrupted by a microfluidizer (MN400BF, Micronox, Seongnam, Republic of Korea) 5 times at 27,000 psi. Bacterial lysates called MSFs that were freeze-dried (PVTFD 20R, iLShinBioBase, Gyeonggi, Republic of Korea). *L. plantarum* K8 lysates were prepared similarly.

### 4.2. Cell Culture and Stimulation

The human keratinocyte (HaCaT) cells were cultured in Dulbecco’s Modified Eagle medium (DMEM), containing 10% fetal bovine serum (FBS) (Thermo Fisher Scientific Solution LLC, Oakville, Canada) and 100 U/mL penicillin (Global Life Sciences Solutions USA LLC, Marlborough, MA, USA) at 37 °C in 5% CO_2_ [20]. The human leukemia monocytic cell line (THP1) cells were maintained in Roswell Park Memorial Institute (RPMI) medium 1640 supplemented with 10% FBS and 1% penicillin–streptomycin at 37 °C in a humidified 5% CO_2_ [21]. For cell tests, 5 × 10^5^ cells were seeded onto a 6-well plate for 24 h, and then the cells were starved for 20 h before pre-treating with different concentrations of MSF or K8 lysate. Similarly, for inhibitor treatment, after starvation, the cells were treated with 10 µM SB203580 (Selleck Chemicals, Houston, TX, USA) or 50 µM SP100030 (Sigma-Aldrich, St. Louis, MO, USA) for 3 h, and then treated with different concentrations of MFS or K8 lysate for 1 h [6].

### 4.3. S. aureus Preparation

Methicillin-resistant *S. aureus* (CCARM 3506) was supplied by the Culture Collection of Antimicrobial Resistant Microbes (CCARM) at Seoul Women’s University in the Republic of Korea. *S. aureus* from glycerol stock was inoculated and grown in tryptic soy broth (TSB) at 37 °C overnight. *S. aureus* were incubated in TSB broth until OD_600_ 0.1; *S. aureus* were killed by heat at 80 °C for 20 min and used for the in vitro experiment. Live *S. aureus* (1 × 10^8^ CFU) were injected into the mice’s ears for in vivo experiment.

### 4.4. Cell Viability

The viability was measured with an MTT (3-(4,5-Dimethylthiazol-2-yl)-2,5-Diphenyltetrazolium Bromide) assay that has been described previously [20]. HaCaT and THP1 cells were seeded at a density of 1 × 10^5^ cells/ well in 96-well plates (BD Biosciences, Franklin Lakes, NJ, USA). Relative cell viability was calculated as a percentage of vehicle-treated control; (OD of treated cells/OD of control × 100).

### 4.5. qRT-PCR Analysis

Total RNA was extracted using a TRIzol reagent (Life Technology, Thermo Fisher Scientific, Waltham, MA, USA). The complementary DNA (cDNA) was synthesized from purified RNA using reverse transcriptase (NanoHelix, Daejeon, Republic of Korea) according to the manufacturer’s instructions. qRT-PCR with 2× real-time PCR master mix (BIOFACT, Daejeon, Republic of Korea) was performed using a CFX96TM Real-time PCR Detection System (Bio-Rad, Hercules, CA, USA) with the recommended universal thermal cycling parameters. Relative gene expression quantification was analyzed using the 2^−ΔΔCq^ method, and the final results were presented as a relative fold change. The expression of glyceraldehyde-3-phosphate dehydrogenase (GAPDH) housekeeping gene served as an internal reference. All assays were performed by duplication on the same plate and were repeated at least three times. The primer sequences used for qRT-PCR are shown in Table 1 [6,20].

### 4.6. Western Blot Analysis

Cells were washed with phosphate-buffered saline (PBS) and lysed in radioimmunoprecipitation assay (RIPA) lysis buffer containing 150 mM sodium chloride, 1% Triton X-100, 0.5% sodium deoxycholate, 0.1% sodium dodecyl sulfate (SDS), 50 mM Tris (pH 8.0), and complete protease inhibitor cocktail (BIOMAX, Gyeonggi, Republic of Korea). Protein concentration was determined by a Brandford (Sigma-Aldrich, St. Louis, MO, USA) assay with bovine serum albumin (BSA) as a standard and detected using a UVITEC imaging system (Uvitec Ltd., Cambridge, UK). Denatured proteins (50 μg) were separated with SDS-PAGE and transferred onto PVDF membranes by a wet transfer apparatus (Bio-Rad, Hercules, CA, USA). Membranes were blocked with 5% BSA (Bioshop, Ontario, Canada) or 5% non-fat milk (Bio-Rad, Hercules, CA, USA) for 1–2 h at room temperature, and then probed with primary antibodies overnight and secondary HRP-conjugated IgG antibody for 1 h at room temperature (Table 2). The protein expression was determined with an ECL reagent (Bio-Rad, Hercules, CA, USA) and detected with a UVITEC imaging system. The results presented are representative of at least three independent experiments.

### 4.7. Animal Experiments

Female BALB/cAnNTac mice (6–8 weeks old) were purchased from JA BIO (Suwon, Republic of Korea). Mice were maintained under standard laboratory conditions (24 ± 2 °C and 50 ± 10% relative humidity). Mice were allowed to acclimate for 1 week before the experiment began. Mice were randomly divided into six groups (Table 3) and three independent experiments were performed [22,23]. At the end of the experiment, mice were sacrificed whose ears were sampled for qRT-PCR.

### 4.8. Statistical Analysis

Data are expressed as the mean ± standard deviations (SD) of three independent experiments. Statistical analysis was performed with GraphPad Prism 5 (GraphPad software). One-way ANOVA with Tukey’s multiple comparisons was used when there were more than two groups. Student’s test was used for normally distributed data analysis. *p*-values of less than 0.05, 0.01, or 0.001 were considered statistically significant.

## 5. Conclusions

Defensins have versatile immune functions that are pivotal for both inhibiting the growth of pathogens and disease progression. This study aimed to provide a platform for designing defensin-based strategies against human microbial pathogens by fermentation products such as MSF. Furthermore, MSF showed strong anti-inflammatory activity that can be used for cosmetic purposes or to develop the lead source of a compound for a new anti-inflammatory medicine.

## Figures and Tables

**Figure 1 molecules-28-02744-f001:**
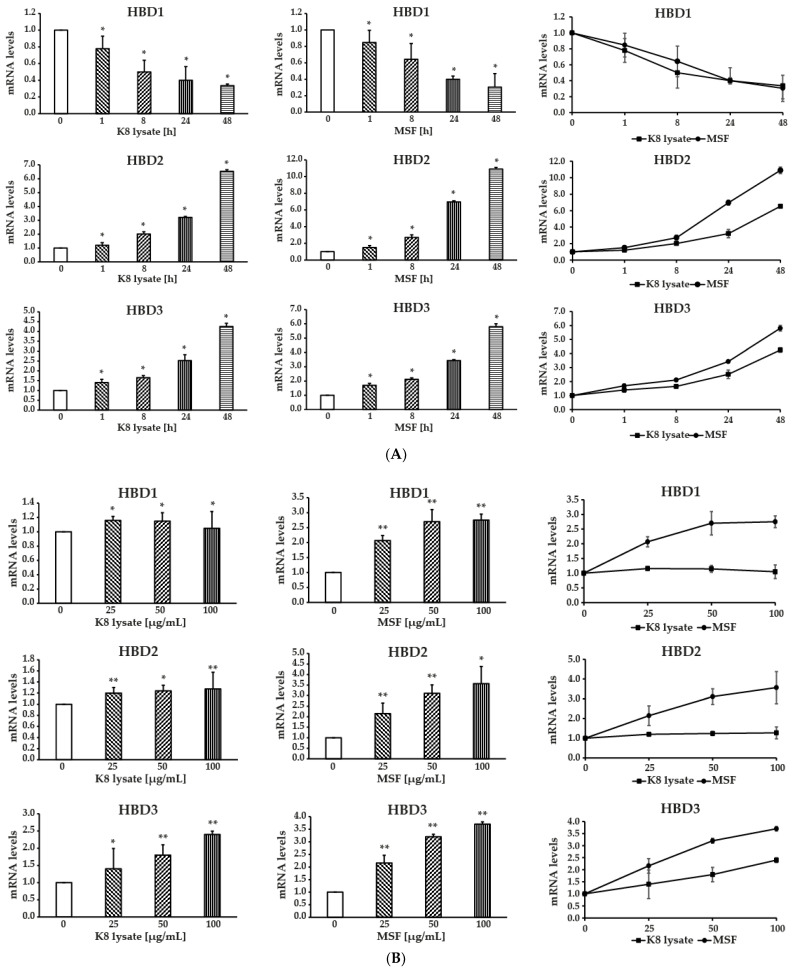
MSF induced the expression of the human antimicrobial peptide human *β*-defensin-2 (HBD2) and *β*-defensin-3 (HBD3). (**A**,**B**) qRT-PCR analysis of HBD1, HBD2, and HBD3 mRNA levels in HaCaT after treatment with MSF or K8 lysate. (**C**) The protein expression of HBD2 and HBD3 was increased by MSF treatment in HaCaT cells for 1 h and Western blot quantification was checked by ImageJ. Values are mean ± SD from three independent experiments. * *p* < 0.05, ** *p* < 0.01.

**Figure 2 molecules-28-02744-f002:**
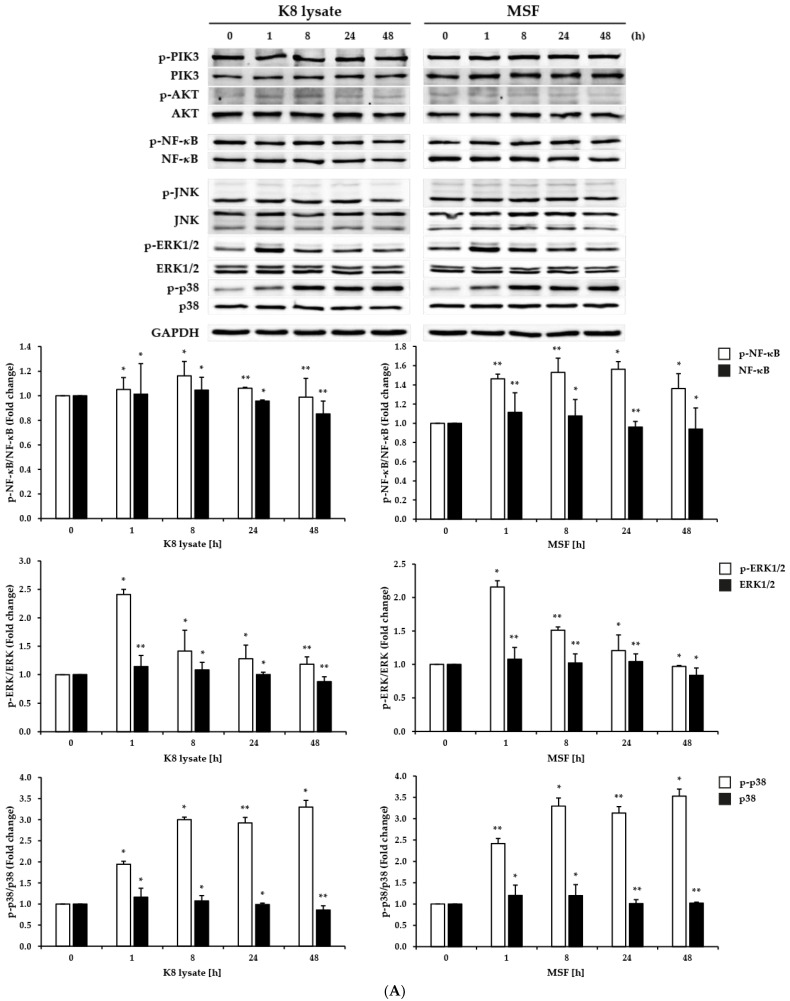
MSF induced the NF-*κ*B and p38 protein levels in HaCaT cells. After 20 h of starvation, HaCaT cells were treated with 25–100 µg/mL of MSF or K8 lysate for 1 h. (**A**,**B**) Western blot analysis of PI3K, AKT, NF-*κ*B, JNK, ERK1/2, and p38 protein levels and Western blot quantification (the white bar is the phosphorylated form, and the black bar is the unphosphorylated form). (**C**) Viability of HaCaT cells after treatment with 1.5625–100 µg/mL of MSF or K8 lysate. Values are mean ± SD from three independent experiments. * *p* < 0.05, ** *p* < 0.01.

**Figure 3 molecules-28-02744-f003:**
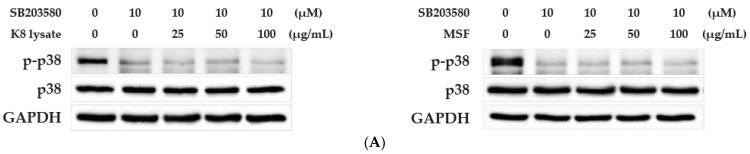
MSF affected the expression of HBD2 and HBD3 through the p38 signaling pathway. HaCaT cells in a confluent cell monolayer were starved for 20 h, and after pre-treatment for 3 h with 10 µM SB203580, were treated with 25–100 µg/mL concentration of MSF or K8 lysate for 1 h. (**A**) Western blot analysis of p38 and the phosphorylated form of p38 in HaCaT cells treated with MSF or K8 lysate in a p38-inhibited condition. (**B**) qRT-PCR reaction of HBD2, and HBD3 expression in HaCaT cells treated with MSF or K8 lysate in a p38-inhibited condition. Values are mean ± SD from three independent experiments. * *p* < 0.05, ** *p* < 0.01.

**Figure 4 molecules-28-02744-f004:**
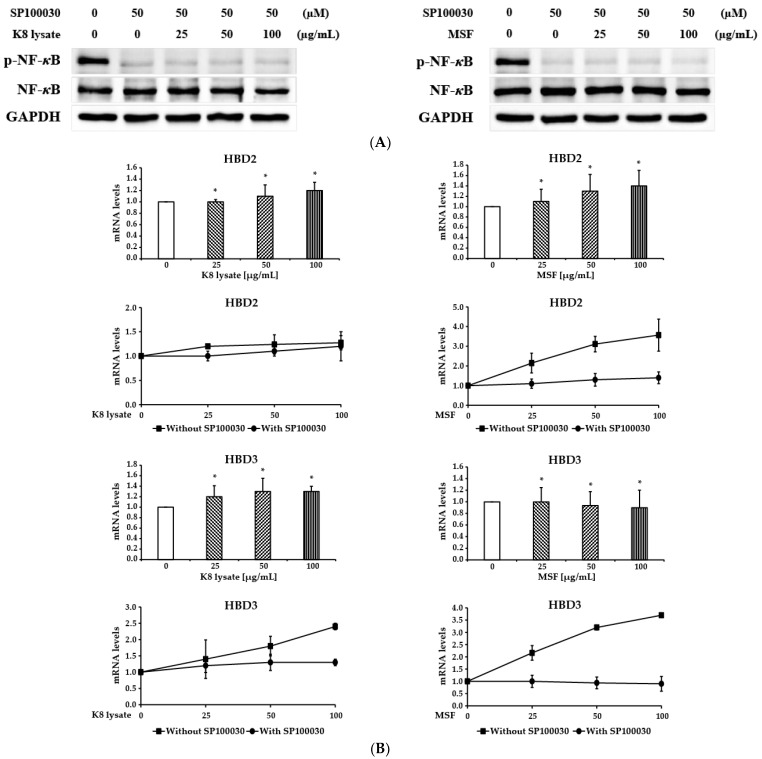
MSF increased the expression of HBD3 through the NF-*κ*B signaling pathway. HaCaT cells were pre-treated with SP100030 (50 µM) for 3 h and then treated with 25 to 100 µg/mL of MSF or K8 lysate for 1 h. (**A**) The expression of NF-*κ*B and the phosphorylated form of NF-*κ*B were detected by Western blot. (**B**) The expressions of HBD2, and HBD3 were checked by qRT-PCR in a NF-*κ*B-inhibited condition. Results are representative of three biological replicates. * *p* < 0.05.

**Figure 5 molecules-28-02744-f005:**
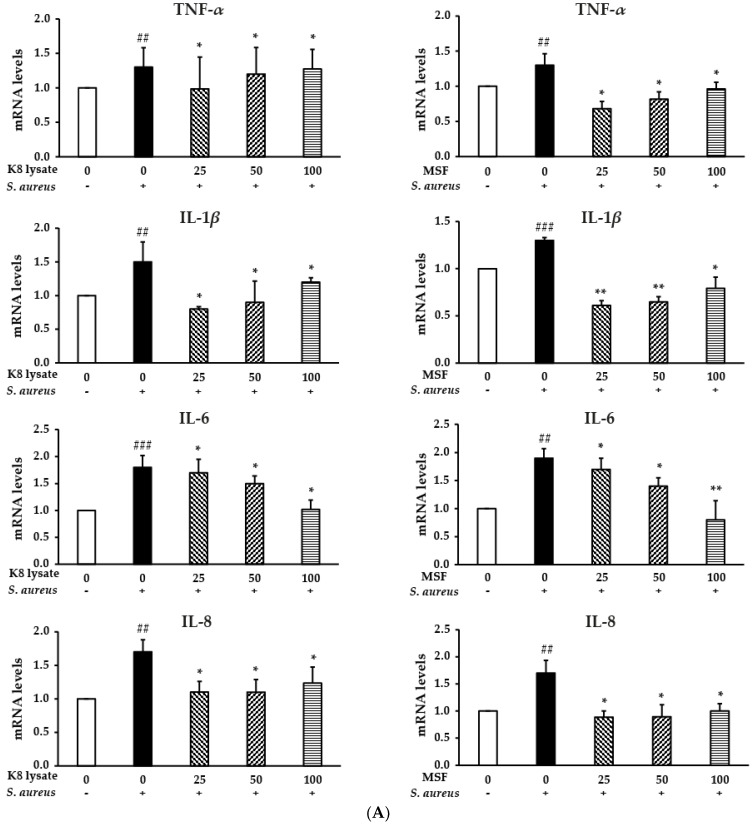
MSF inhibited the *S. aureus*-induced inflammation in THP1 cells. THP1 cells were pre-treated with MSF or K8 lysate (25, 50, and 100 µg/mL) for 1 h and then stimulated with heat-killed *S. aureus* (1 × 10^8^ CFU) for 24 h. (**A**) The expression of TNF-*α*, IL-1*β*, IL-6, and IL-8 was detected by qRT-PCR. (**B**) The protein levels of MAPKs and NF-*κ*B were checked by Western blot and Western blot quantification (the white bar is the phosphorylated form, and the black bar is the unphosphorylated form). (**C**) Cell viability of THP1 cells was tested by MTT assay with the indicated concentrations of MSF or K8 lysate. Data represent mean ± SD from three independent experiments. ^#^ *p* < 0.05, ^##^ *p* < 0.01, ^###^ *p* < 0.001 vs. control. * *p* < 0.05, ** *p* < 0.01 vs. model.

**Figure 6 molecules-28-02744-f006:**
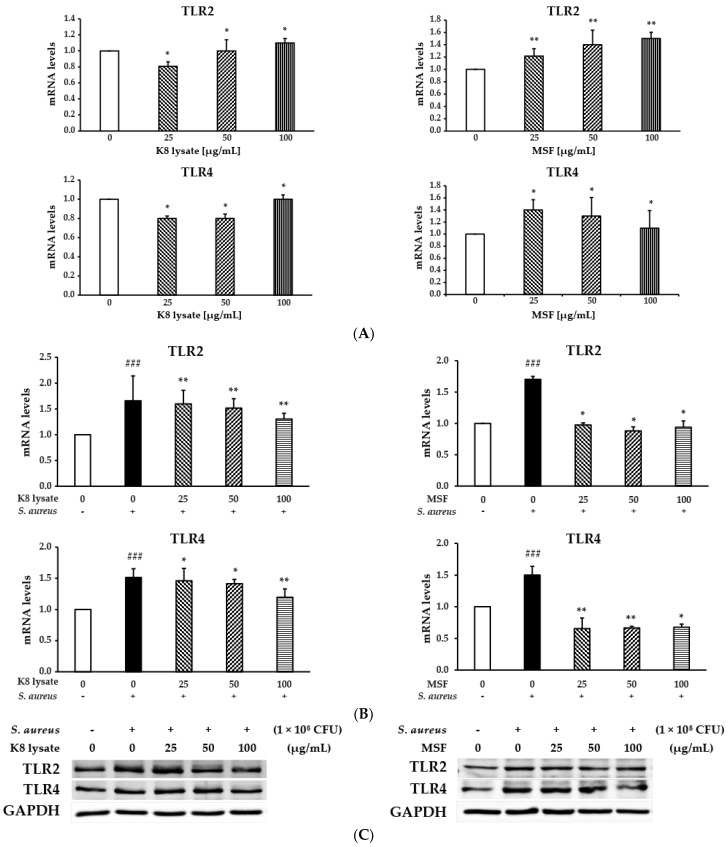
MSF treatment inhibited the expression of TLR2 in HaCaT and THP1 cells. (**A**) qRT-PCR analysis of TLR2 and TLR4 in HaCaT cells treated with 25–100 µg/mL of MSF or K8 lysate. (**B**) qRT-PCR analysis of TLR2 and TLR4 in THP1 cells stimulated by *S. aureus* treated with MSF or K8 lysate. (**C**) Western blot analysis. Data represent mean ± SD from three independent experiments. ^###^ *p* < 0.001 vs. control. * *p* < 0.05, ** *p* < 0.01 vs. model.

**Figure 7 molecules-28-02744-f007:**
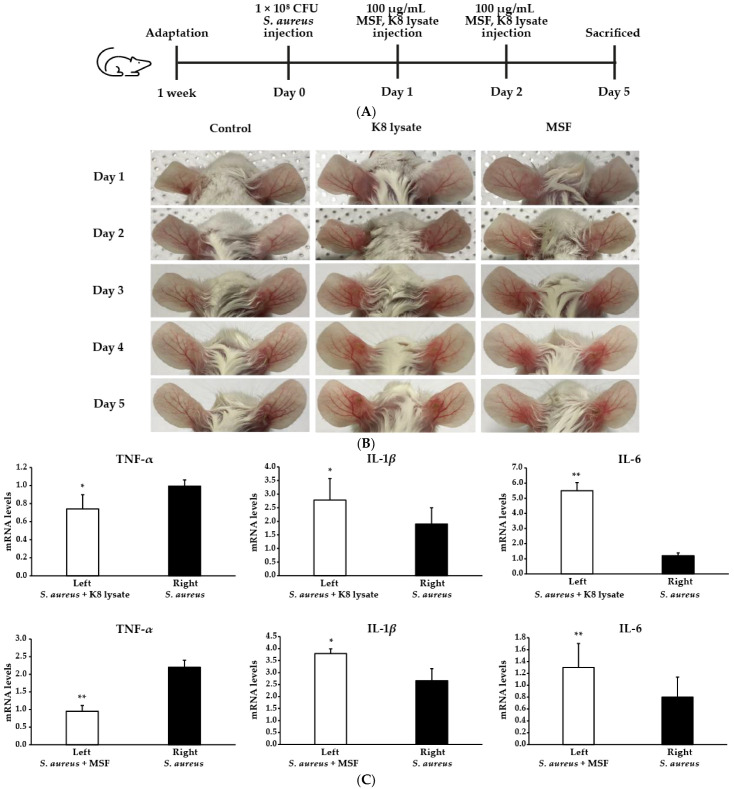
MSF reduced inflammation in an *S. aureus*-injected ears of a mouse model. Live *S. aureus* (1 × 10^8^ CFU) were inoculated into the ear of mice together with or without MSF or K8 lysate (100 µg/mL). (**A**) Experimental scheme for in vivo studies. (**B**) Ears with an inflammatory phenotype. (**C**) The mRNA of TNF-*α*, IL-1*β*, and IL-6 expression levels were measured from the ears tissues using qRT-PCR. * *p* < 0.05, ** *p* < 0.01 vs. right mouse ear.

**Figure 8 molecules-28-02744-f008:**
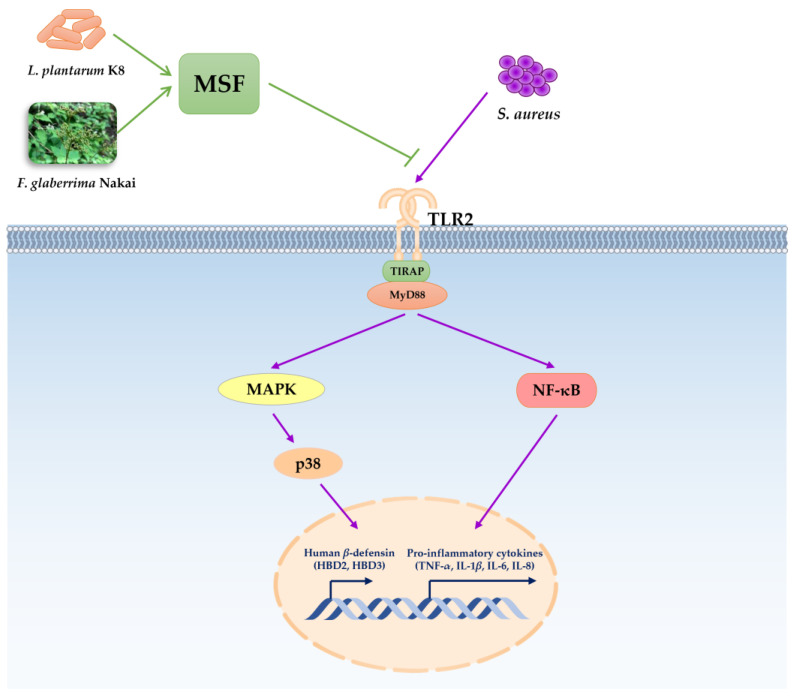
Schematic diagram illustrating the proposed mechanism by which MSF induces the expression of HBD2, and HBD3 and suppresses *S. aureus*-induced inflammatory responses by activating TLR2 through the p38 and NF-*κ*B signaling pathways.

**Table 1 molecules-28-02744-t001:** Primers used for qRT-PCR analysis.

Primers	Forward	Reverse
hBD1	GTCGCCATGAGAACTTCCTACC	CATTGCCCTCCACTGCTGAC
hBD2	CCTGTTACCTGCCTTAAGAGTG	GAATCCGCATCAGCCACAG
hBD3	CTTCTGTTTGCTTTGCTCTTCC	CACTTGCCGATCTGTTCCTC
hTNF-*α*	TGAGCACTGAAAGCATGATCC	ATCACTCCAAAGTGCAGCAG
hIL-1*β*	TCTTTGAAGAAGAGCCCGTCCTC	GGATCCACACTCTCCAGCTGCA
hIL-6	CCTGAACCTTCCAAAGATGGC	CACCAGGCAAGTCTCCTCATT
hIL-8	TCTGTGTGAAGGTGCAGTTTT	GGGGTGGAAAGGTTTGGAGTA
hTLR2	TGTCTTGTGACCGCAATGGT	GTTGGACAGGTCAAGGCTTT
hTLR4	CCCTGAGGCATTTAGGCAGCTA	AGGTAGAGAGGTGGCTTAGGCT
hGAPDH	GTGAAGGTCGGAGTCAACG	TGAGGTCAATGAAGGGGTC

**Table 2 molecules-28-02744-t002:** Antibodies used for Western blot analysis.

Antibody	Source	Category No.
p-PI3K	Cell Signaling Technology	4257S
PI3K	Cell Signaling Technology	4228S
p-AKT	Cell Signaling Technology	9271S
AKT	Cell Signaling Technology	9272S
p-NF-*κ*B	Cell Signaling Technology	3033S
NF-*κ*B	Santa Cruz Biotechnology	sc-8008
p-p38	Cell Signaling Technology	9211S
p38	Santa Cruz Biotechnology	sc-535
p-ERK1/2	Cell Signaling Technology	9101S
ERK1/2	Cell Signaling Technology	9102S
p-JNK	Santa Cruz Biotechnology	sc-6254
JNK	Santa Cruz Biotechnology	sc-7345
HBD2	Abcam	ab9871
HBD3	Abcam	ab172703
GAPDH	Cell Signaling Technology	14C10

**Table 3 molecules-28-02744-t003:** The group of in vivo experiments.

	Left Ear		Right Ear	
	Compound	Pathogen	Compound	Pathogen
**Group 1**	None	None	PBS	None
**Group 2**	None	None	None	*S. aureus*
**Group 3**	K8 lysate	None	PBS	None
**Group 4**	K8 lysate	*S. aureus*	PBS	*S. aureus*
**Group 5**	MSF	None	PBS	None
**Group 6**	MSF	*S. aureus*	PBS	*S. aureus*

## Data Availability

Not applicable.

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
