# Peer review of "MSF Enhances Human Antimicrobial Peptide β-Defensin (HBD2 and HBD3) Expression and Attenuates Inflammation via the NF-κB and p38 Signaling Pathways"

_molecules, 2023, doi:10.3390/molecules28062744_

Round 1
Reviewer 1 Report (Previous Reviewer 3)
I still have concerns with the choice of inhibitor used in Figure 4. In the text line 167 the authors state LY294002 is a specific NFKB inhibitor when it fact its a PI3K inhibitor. This is a confusing method to inhibit NFKB when there are more specific inhibitors such as Withaferin A.
I believe the authors need a different approach to inhibiting NFKB for the paper to be considered for publication.
Author Response
Please see the attachment.

Reviewer 2 Report (New Reviewer)
This manuscript describes MSF enhances human antimicrobial peptide β-defensin expression and attenuates inflammation via the NF-κB and p38 signaling pathways. The author evaluated the effect of MSF through in vitro and in vivo experiments, and verified the results through inhibitors. But there are some data that are not reasonable and it is suggested that the authors should modify and supplement and further analyze data. It is not suitable for publishing now and it is recommended that the author major revision. After careful review, some comments are provided below.
Major comments:
1. In Result 2.1, whether the author controls other variables to remain unchanged and compares MSF and K8, otherwise the results of this section have errors. It is suggested that the author optimize the WB result in Figure 1C, the band background is too dark, and the protein molecular weight needs to be marked.
2. In Figure 2B, the expression of p-p38 in MSF increases with the increase of dose, while in Figure 3A, the author needs to explain why the expression of p-p38 in MSF decreases with the increase of dose after the use of p38 inhibitor.
3. In Result 2.3, the author compares the results in Figure 3B with those in Figure 1B and draws a conclusion, which is inappropriate. The above problems also exist in Result 2.4.
4. In Figure 5A, IL-6 of 25ug/ml MSF should not be marked with significance.
5. Why IL-6 and IL-1β in the ears of mice treated with MSF is higher than that in the other ear.
6. The author needs to redraw Figure 8, some parts of the figure have not been verified in the above and are not recommended to appear in this figure.
Round 2
Reviewer 1 Report (Previous Reviewer 3)
The addition of the data with the more specific NFKB inhibitor addresses my major concern regarding the manuscript. I now believe the manuscript is suitable for publication.
Author Response
Please see the attachment.

This manuscript is a resubmission of an earlier submission. The following is a list of the peer review reports and author responses from that submission.
Round 1
Reviewer 1 Report
The gray value of WB strip should be quantified
Inflammatory factors increase in serum but their mRNA expression decreases, which is not explained clearly in the discussion part.
Reviewer 2 Report
In the present manuscript, the authors analysed if the addition of F. glaberrima extract in the growth medium of L. plantarum K8 (MSF) affects some immunomodulatory properties of the bacterial lysate both in in vitro and in vivo models. The authors showed that MSF treatment increases defensin production and modulates pro-inflammatory cytokine expression in HaCat cells through NF-kB and P38 activation. Moreover, MSF treatment was shown to reduce the inflammatory response in THP1 cell exposed to heat-killed S aureus. MSF was shown to modulate TLR2 and 4 expression in both THP1 and HaCat cells. Finally, MSF treatment was shown to reduce the inflammatory response in a murine model of ear inflammation.
Overall, the manuscript is quite confused and several items need to be addressed.
Methods: with the exception of some WB data, it is not specified which dose of K8 or MSF has been used in time-course experiments and which time-point in dose –curve experiments.
Statistic approach is not correct. When more than 2 comparisons have to be performed, one-way or two-way ANOVA followed by post-hoc test should be used. Moreover, to claim the time- or dose-dependence of a treatment, a statistical significant difference has to be demonstrated among the different time-points or doses. This applies to all the results presented in all the figures.
As concerns results in fig 1 about the protein expression of defensins, 1h treatment seems a very short time to see the increase in protein levels especially according to mRNA data. Moreover, WB data have to be quantified and the relevant statistical analysis performed.
As concerns fig2, due to the very limited variations in the levels of mRNA expression of most of the analysed cytokines, mRNA data for pro-inflammatory cytokines should be confirmed at protein levels. Moreover, since the difference in the kinetics of TNF (climax at 48h) and the other cytokines (climax at 1h), was the dose-curve analysis performed at the same time-point? In the description of the results (lanes 111-112) it does not seem that IL1, IL6 and IL8 are time-dependently reduced but that after an initial increase the mRNA levels return toward base-line values. Protein level determination may solve this misinterpretation.
Data in fig 3A and 4A should be quantified and results statistically analysed. The effect of P38 and NFkB inhibition should be directly compared with the effect in the absence of the inhibitors (Fig 3B and C and Fig 4B and C.
Fig5. The authors should demonstrate that S aureus treatment actually induces the analysed genes before treatment and protein determination should be also performed.
Fig 6 TLR protein levels or membrane expression should be analysed. Again, before treating with K8 and MSF, the authors should demonstrate that TLR2 and TLR4 are actually induced (i.e. they are statistically different from untreated controls). Moreover, since MSF reduces TLR2 expression and S.Aureus induces it, how can the author explain that the combined treatment leads to decrease in TLR2 expression? Protein data could help to better clarify this aspect.
Fig7 the images of the ears are unclear. The authors should quantify the inflammatory effect (thickness of the ear, amount of vascularization).
Discussion: The authors should better explain the experimental bases of their hypothesis that MSF reduces S Aureus viability.
Fig 8 is unclear. It does not show the possible levels of interference of MSF on S Aureus activation pathway.
Iconography: As a general consideration, iconography has to be ameliorated. Fig 1 is redundant. Only the panels on the right can be shown, without any loss of information. Fig2 WB images are confounding, only the relevant determinations should be shown, not affected pathways can be shown as supplementary data or indicated as data not shown. Data in fig 2C and D could be summarized as in the right panels in fig 1. This way to represent the data could also highlights the differences in the strength of the induction that, in the actual view, is flattened by the use of a different scale in the graphs. Graph depicted in fig 3 are counterintuitive. To show the effect of an inhibitor, the authors should compare, on the same graph, the effect in the presence or in the absence of the inhibitor. For fig 3 and 4, 5A,5C, and 6, the same advices for the previous figure can be applied, i.e. merge the graphs of the two treatments (MSF and K8) to allow for a direct comparison. For fig 2B see the comment for fig2A and B.
Reviewer 3 Report
In this study Nguyen et al, analysis the effects of the probiotic lysates MSF and K8 on the expression of human defensins in skin keratinocytes. They show to varying degrees that the lysates can induce expression of the denfensins and this is induced partially through the p38 signalling pathway with the NFKB pathway playing a role in HBD3 specifically. Finally they show the lysates have a anti-inflammatory effect on THP1 cells exposed to S.aureus
The manuscript requires some more details in a number of sections and the inductions of the signalling pathways is small therefore it requires quantitive analysis. Below are specific comments
In the introduction more details about MSF should be included including functions etc. The authors describe L.Plantarum as a probiotic. Therefore the authors should include a rational for looking at it in the skin within the introduction.
In the methods the authors should describe the method of purifying MSF and K8 from the cultures.
In figure 1C could the authors perform densitometry analysis of the blots
In section 2.2 The authors should explain the rational for looking at the signalling pathways described.
In figure 2A and B the activation of NFKB and pERK is minimal. The authors meed to perform quantitive analysis of the blots.
Figure 3 and 4 The authors need to include K8 and MSF stimulation without the p38 and NFKB inhibitors in the same experimental conditions.
Section 2.5 needs a introduction as to why you started to look at MSF/K8 on THP1 cells stimulated with S aureus.
Round 2
Reviewer 2 Report
Revised manuscript is not improved. Statistical analysis is unclear. It seems that untreated controls are all set=0 for all the replicates. This means that control cells do not have any variance. How did the authors perform the anova test? Moreover, the authors have to declare which comparisons have been performed in the statistical analysis. In figure 1A and B, if the authors want to stress the difference between K8 and MSF a statistical test should be performed. In fig. 2A from the densitometric analysis t is very hard to appreciate the statistical difference that is indicated at all the analysed times. Moreover, the identity of black and white bars are not detailed. In figure 3A it seems that in untreated HACAT cells p38 is already heavily phosphorylated . This is in contrast with previous WB data. Overall, the manuscript is very confused and no effort seems to have been done to ameliorate the iconography.
Reviewer 3 Report
The authors have improved the manuscript with the additonal information and data. There are a few outstanding issues.
In the densitometry analysis graphs in figure 2. I assume the white bars are for phospho protein levels and the black bars represent total protein levels. There is no key or explanation of the different bars in the legend.
In figure 4 It looks like the NFKB inhibitor LY294002 Does not work in the experiments. There is no reduction in pNFKB levels in the western blots when the compound is added. This suggests the effects seen in HBD2/3 levels is due to off target effects of the inhibitor.
